# Predicting the effect of statins on cancer risk using genetic variants from a Mendelian randomization study in the UK Biobank

Paul Carter[1], Mathew Vithayathil[2], Siddhartha Kar[3,4], Rahul Potluri[5], Amy M Mason[1], Susanna C Larsson[6,7], Stephen Burgess[1,8]*

[1]Department of Public Health and Primary Care, University of Cambridge, Cambridge, United Kingdom; [2]MRC Cancer Unit, University of Cambridge, Cambridge, United Kingdom; [3]MRC Integrative Epidemiology Unit, University of Bristol, Bristol, United Kingdom; [4]Population Health Sciences, Bristol Medical School, University of Bristol, Bristol, United Kingdom; [5]ACALM Study Unit, Aston Medical School, Birmingham, United Kingdom; [6]Unit of Cardiovascular and Nutritional Epidemiology, Institute of Environmental Medicine, Karolinska Institutet, Stockholm, Sweden; [7]Department of Surgical Sciences, Uppsala University, Uppsala, Sweden; [8]MRC Biostatistics Unit, University of Cambridge, Cambridge, United Kingdom

**Abstract** Laboratory studies have suggested oncogenic roles of lipids, as well as anticarcinogenic effects of statins. Here we assess the potential effect of statin therapy on cancer risk using evidence from human genetics. We obtained associations of lipid-related genetic variants with the risk of overall and 22 site-specific cancers for 367,703 individuals in the UK Biobank. In total, 75,037 individuals had a cancer event. Variants in the *HMGCR* gene region, which represent proxies for statin treatment, were associated with overall cancer risk (odds ratio [OR] per one standard deviation decrease in low-density lipoprotein [LDL] cholesterol 0.76, 95% confidence interval [CI] 0.65–0.88, p=0.0003) but variants in gene regions representing alternative lipid-lowering treatment targets (*PCSK9*, *LDLR*, *NPC1L1*, *APOC3*, *LPL*) were not. Genetically predicted LDL-cholesterol was not associated with overall cancer risk (OR per standard deviation increase 1.01, 95% CI 0.98–1.05, p=0.50). Our results predict that statins reduce cancer risk but other lipid-lowering treatments do not. This suggests that statins reduce cancer risk through a cholesterol independent pathway.

*For correspondence:
sb452@medschl.cam.ac.uk

Competing interests: The authors declare that no competing interests exist.

## Introduction

Statins are inhibitors of 3-hydroxy-3-methyl-glutaryl-coenzyme A reductase (HMGCR), which is the rate-limiting enzyme in the mevalonate pathway; a pathway producing a range of cell signaling molecules with the potential to regulate oncogenesis. This is supported by strong laboratory evidence that statins induce anticarcinogenic effects on cell proliferation and survival in various cell lines (*Sławińska-Brych et al., 2014*; *Crosbie et al., 2013*; *Ishikawa et al., 2014*; *Chang et al., 2013*), and reduce tumor growth in a range of in vivo models (*Narisawa et al., 1996a*; *Inano et al., 1997*; *Narisawa et al., 1996b*; *Clutterbuck et al., 1998*; *Kikuchi et al., 1997*; *Hawk et al., 1996*). Furthermore, epidemiological studies of pre-diagnostic use of statins have been associated with reduced risk of specific cancer types (*Khurana et al., 2007*; *Poynter et al., 2005*; *Pocobelli et al., 2008*). However, meta-analyses of cardiovascular-focused randomized controlled trials have shown no effect

of statins on cancer (*Dale et al., 2006*; *Farooqi et al., 2018*). Conclusions from these trials are limited as they lack adequate power and longitudinal follow-up necessary for assessing the impact on cancer risk. At present, no clinical trials have been designed to assess the role of statins in primary cancer prevention and their role in chemoprevention remains uncertain.

A putative protective effect of statins on cancer development could be through either cholesterol-dependent or independent effects (*Goldstein and Brown, 1990*; *Mullen et al., 2016*; *Yeganeh et al., 2014*; *Denoyelle et al., 2001*). Cholesterol is a key mediator produced by the mevalonate pathway and is essential to cell signaling and membrane structure, with evidence demonstrating the potential to drive oncogenic processes and tumor growth (*Chen and Resh, 2002*; *Li et al., 2006*). However, the epidemiological relationships between circulating cholesterol and cancer risk remain unclear. Individual observational studies have reported positive (*Kitahara et al., 2011*; *Strohmaier et al., 2013*), inverse (*Kitahara et al., 2011*; *Strohmaier et al., 2013*; *Melvin et al., 2012*; *Katzke et al., 2017*), and no association (*Salonen, 1982*; *JPHC Study Group et al., 2009*; *Van Hemelrijck et al., 2012*; *Ma et al., 2016*) between circulating levels of total cholesterol, low-density lipoprotein (LDL) cholesterol, high-density lipoprotein (HDL) cholesterol, and triglycerides with the risk of overall and site-specific cancers. Different cancer types have distinct underlying pathophysiology, and meta-analyses of observational studies highlight a likely complex relationship which varies according to both lipid fraction (*Vílchez et al., 2014*; *Alsheikh-Ali et al., 2008*) and cancer type (*Ma et al., 2016*; *Lin et al., 2017*; *Passarelli and Newcomb, 2016*; *Yao and Tian, 2015*). Furthermore, cancer can lower cholesterol levels for up to 20 months before diagnosis (*Kritchevsky et al., 1991*). Thus, the true relationship between lipids and cancer development remains equivocal.

Mendelian randomization is an epidemiological approach that assesses associations between genetically predicted levels of a risk factor and a disease outcome to predict the causal effect of the risk factor on an outcome (*Davey Smith and Hemani, 2014*). The use of genetic variants minimizes the influence of reverse causality and confounding factors on estimates. Mendelian randomization studies also have the potential to predict the outcomes of trials for specific therapeutic interventions. A limited number of Mendelian randomization studies have investigated the relationship between HMGCR inhibition and cancer (*PRACTICAL consortium et al., 2016*; *Rodriguez-Broadbent et al., 2017*; *Orho-Melander et al., 2018*; *Nowak and Ärnlöv, 2018*; *Yarmolinsky et al., 2020*), with protective associations observed for prostate cancer (*PRACTICAL consortium et al., 2016*), colorectal cancer (*Rodriguez-Broadbent et al., 2017*), breast cancer (*Orho-Melander et al., 2018*; *Nowak and Ärnlöv, 2018*), and ovarian cancer (*Yarmolinsky et al., 2020*). However, no comprehensive Mendelian randomization investigation has evaluated the predicted impact of HMGCR inhibition or the causal role of specific lipid fractions on the risk of many of the most common site-specific cancers.

Here we investigate the relationship between HMGCR inhibition and the risk of overall cancer and site-specific cancers using genetic variants in the *HMGCR* gene region. To understand whether statins may influence cancer risk through lipid-related mechanisms, we also assess the relationship between lipids and cancer risk by polygenic Mendelian randomization analyses using common lipid-associated genetic variants. Additionally, to mimic other lipid-lowering pharmaceutical interventions, gene-specific analyses were performed using variants in or near gene regions targeted by these therapies.

## Results

### Participant characteristics and power calculations

Baseline characteristics of the participants in the UK Biobank and numbers of outcomes are provided in *Table 1*. In total, 75,037 of the participants had a cancer event, of which 48,674 participants had one of the 22 defined site-specific cancers. Power calculations for the various analyses are presented in *Figure 1* (site-specific cancers) and *Supplementary file 1* (overall cancer). The number of cases ranged from 324 for liver cancer to 13,666 for breast cancer with an overall median number of 1462 cases across cancer sites. Gene-specific analyses were only well-powered for overall cancer. Polygenic analyses were well-powered to detect moderate effects for overall cancer and common site-specific cancers but less well-powered for less common site-specific cancers.

**Table 1.** Baseline characteristics of the UK Biobank participants included in this study and the numbers of outcome events.

| Characteristic or cancer site/type | Mean (SD) or N (%)[†] |
| --- | --- |
| Sample size | 367,703 (100) |
| Female | 198,904 (54.1) |
| Age at baseline | 57.2 (8.1) |
| Body mass index | 27.3 (4.8) |
| Systolic blood pressure | 137.6 (18.6) |
| Diastolic blood pressure | 82.0 (10.1) |
| Smoking status (current/ex/ never)* | 37,866 (10.3)/185,704 (50.5)/143,777 (39.1) |
| Alcohol status (current/ex/ never)* | 342,797 (93.2)/12,732 (3.5)/11,646 (3.2) |
| History of type 2 diabetes | 15,834 (4.3) |
| Overall cancer | 75,037 (20.4) |
| Breast | 13,666 (6.9) |
| Prostate | 7872 (4.7) |
| Lung | 2838 (0.8) |
| Bowel | 5486 (1.5) |
| Melanoma | 4869 (1.3) |
| Non-Hodgkin's lymphoma | 2296 (0.6) |
| Kidney | 1310 (0.4) |
| Head/neck | 1615 (0.4) |
| Brain | 810 (0.2) |
| Bladder | 2588 (0.7) |
| Pancreas | 1264 (0.3) |
| Uterus | 1931 (1.0) |
| Leukaemia | 1403 (0.4) |
| Esophagus | 843 (0.2) |
| Ovaries | 1520 (0.8) |
| Gastric | 736 (0.2) |
| Liver | 324 (0.1) |
| Myeloma | 656 (0.2) |
| Thyroid | 375 (0.1) |
| Biliary | 387 (0.1) |
| Cervix | 1928 (1.0) |
| Testes | 735 (0.4) |

*Excluding 356 participants with smoking status absent and 528 participants with alcohol consumption status absent.
[†]For sex-specific cancers, this is the percentage of individuals of the relevant sex.

## Gene-specific analyses for HMGCR and other drug proxy variants

Associations for specific gene regions representing targets of lipid-lowering drugs are displayed in *Figure 2* and *Figure 2—figure supplements 1–6*. For overall cancer, there was evidence of association for variants in the *HMGCR* gene region (odds ratio [OR] 1.32, 95% confidence interval [CI] 1.13–1.53, p=0.0003) but not for other gene regions: for *PCSK9* (OR 1.03, 95% CI 0.92–1.14, p=0.66), for *LDLR* (OR 0.99, 95% CI 0.92–1.07, p=0.86), for *NPC1L1* (OR 0.87, 95% CI 0.73–1.04, p=0.13), for *APOC3* (OR 1.08, 95% CI 0.98–1.19, p=0.15), or for *LPL* (OR 1.03, 95% CI 0.95–1.13, p=0.45). The association of variants in the *HMGCR* gene region with overall cancer remained broadly similar when restricting outcomes to the 48,674 individuals who had one of the 22 site-specific cancers (OR 1.29,

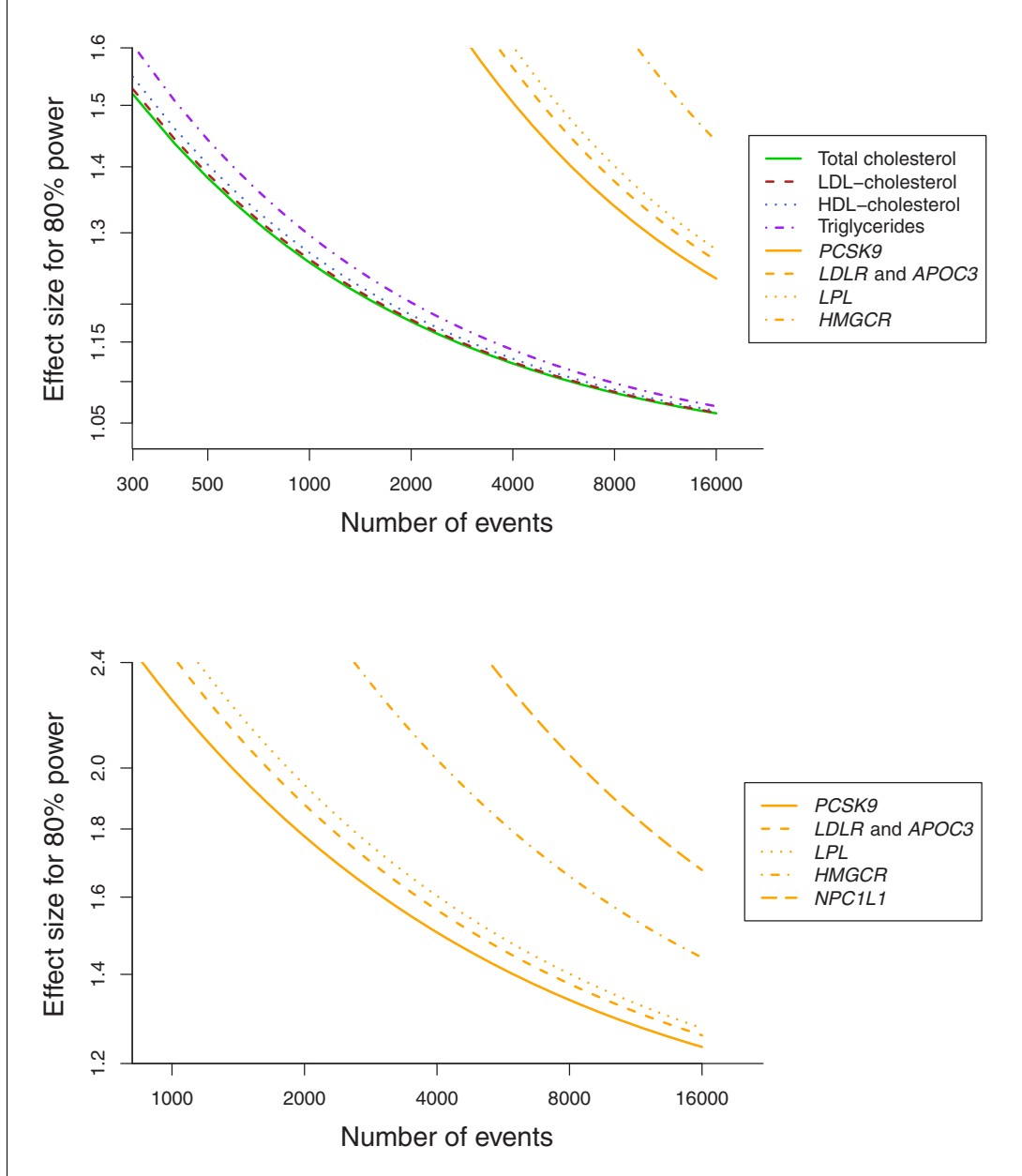

**Figure 1.** Power calculations for polygenic and gene-specific analyses, displaying the Mendelian randomization estimate that can be detected with 80% power assuming a sample size of 367,703 individuals for site-specific cancers.

95% CI 1.08–1.54, p=0.005) and when excluding outcomes that were self-reported only (70,734 remaining cases, OR 1.30, 95% CI 1.12–1.52, p=0.0007).

For site-specific cancers, the *HMGCR* gene region showed positive associations for five of the six most common cancer sites (breast, prostate, melanoma, lung, and bladder; not for bowel), although none of these results individually reached a conventional level of statistical significance. Similar results were observed for analyses of site-specific cancers when excluding individuals with solely self-reported outcomes from the analysis (*Figure 2—figure supplement 7*) and when individuals with a cancer diagnosis other than the site-specific cancer under analysis were omitted from the analysis rather than treated as a control (*Figure 2—figure supplement 8*); estimates were generally slightly higher, but findings were unchanged. There was little evidence for associations in site-specific analyses for other lipid-lowering drug targets.

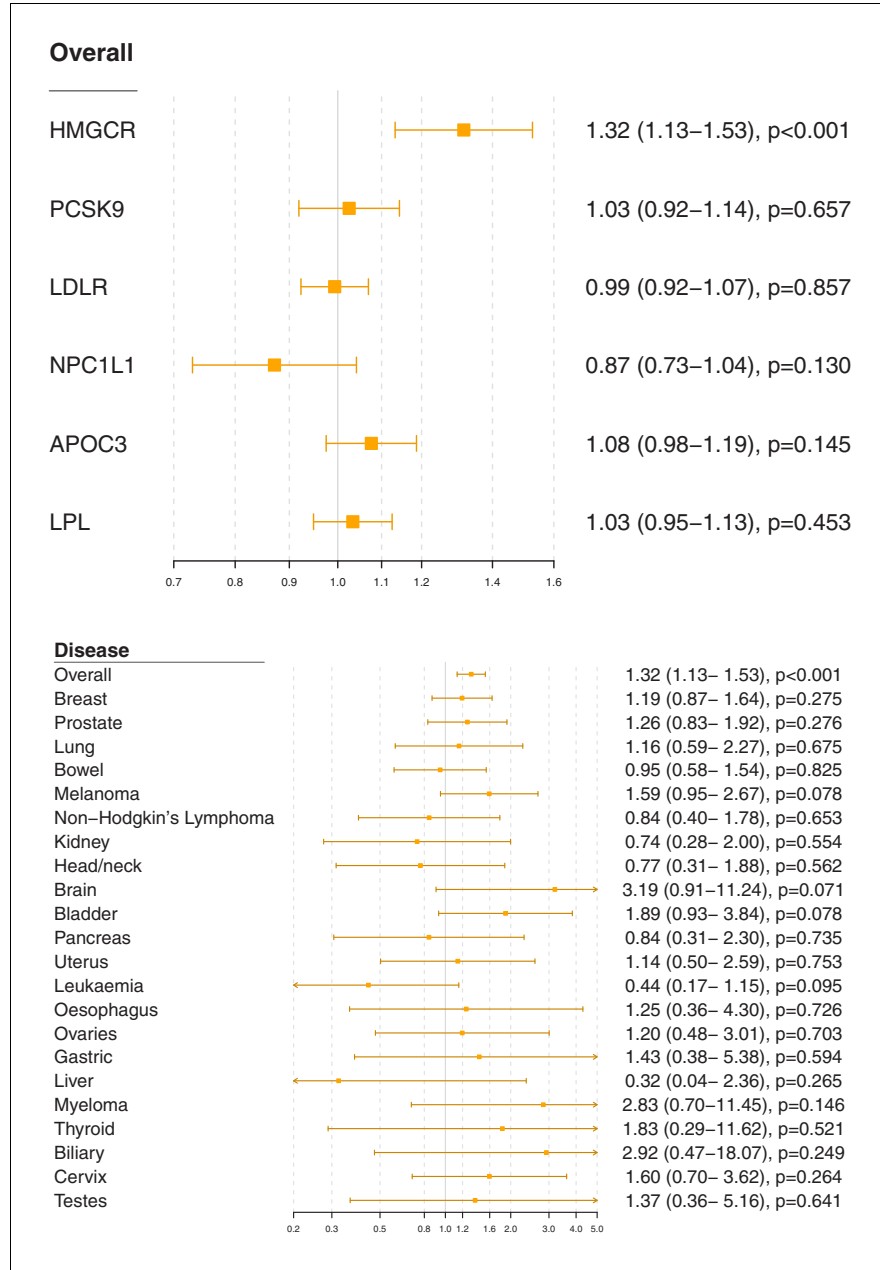

**Figure 2.** Gene-specific Mendelian randomization estimates (odds ratio with 95% confidence interval per one standard deviation increase in lipid fraction) for variants in gene regions representing targets of lipid-lowering treatments. Estimates are scaled to a one standard deviation increase in LDL-cholesterol for the *HMGCR*, *PCSK9*, *LDLR,* and *NPC1L1* regions, and a one standard deviation increase in triglycerides for the *APOC3* and *LPL* regions. A: associations with overall cancer for each gene region in turn. B: associations with site-specific cancers for variants in the *HMGCR* gene region.

The online version of this article includes the following figure supplement(s) for figure 2:

**Figure supplement 1.** Gene-specific Mendelian randomization estimates (odds ratio with 95% confidence interval per one standard deviation increase in LDL-cholesterol) for variants in the *PCSK9* gene region.

**Figure supplement 2.** Gene-specific Mendelian randomization estimates (odds ratio with95%confidence interval per one standard deviation increase in LDL-cholesterol) for variants in the*LDLR*gene region.

**Figure supplement 3.** Gene-specific Mendelian randomization estimates (odds ratio with 95% confidence interval per one standard deviation increase in LDL-cholesterol) for variants in the*NPC1L1*gene region.

**Figure supplement 4.** Gene-specific Mendelian randomization estimates (odds ratio with 95% confidence interval per one standard deviation increase in LDL-cholesterol) for variants in the*APOC3*gene region.

*Figure 2 continued on next page*

*Figure 2 continued*

**Figure supplement 5.** Gene-specific Mendelian randomization estimates (odds ratio with 95% confidence interval per one standard deviation increase in LDL-cholesterol) for variants in the *LPL* gene region.

**Figure supplement 6.** Genetic associations with LDL-cholesterol (standard deviation units) plotted against genetic associations with overall cancer (log odds ratios) for six variants in the *HMGCR* gene region.

**Figure supplement 7.** Gene-specific Mendelian randomization estimates (odds ratio with 95% confidence interval per one standard deviation increase in LDL-cholesterol) for variants in the *HMGCR* gene region excluding self-reported outcomes.

**Figure supplement 8.** Gene-specific Mendelian randomization estimates (odds ratio with 95% confidence interval per one standard deviation increase in LDL-cholesterol) for variants in the *HMGCR* gene region excluding those with a cancer diagnosis other than site-specific cancer under analysis.

## Polygenic analyses for all lipid-related variants

Polygenic Mendelian randomization estimates are displayed in *Figure 3* for HDL-cholesterol, LDL-cholesterol, and triglycerides (see also *Supplementary file 1*), and *Figure 4* for total cholesterol. For overall cancer, the OR per one standard deviation increase in genetically-predicted levels of the risk factor was 1.01 (95% CI 0.98–1.05, p=0.50) for LDL-cholesterol, 0.99 (95% CI 0.95–1.03, p=0.54) for HDL-cholesterol, 1.00 (95% CI 0.95–1.05, p=0.85) for triglycerides, and 1.01 (95% CI 0.98–1.05; p=0.57) for total cholesterol. Results for the lipid subfractions were similar using the multivariable MR-Egger method (*Supplementary file 1*). Similar results were observed when omitting self-reported outcomes from the analysis (*Supplementary file 1*).

For site-specific cancers, there were positive associations between risk of bowel cancer and genetically predicted levels of total cholesterol (OR 1.18, 95% CI 1.06–1.32, p=0.002) and LDL-cholesterol (OR 1.16, 95% CI 1.04–1.29, p=0.006). Compared to primary analyses, results were attenuated in robust methods (*Supplementary file 1*). While these robust methods are univariable

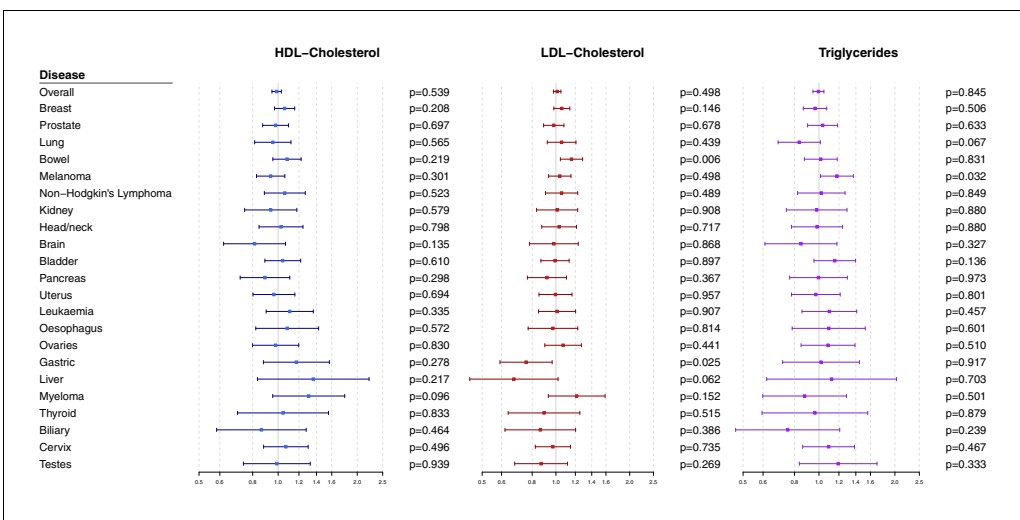

**Figure 3.** Multivariable Mendelian randomization estimates for HDL-cholesterol, LDL-cholesterol, and triglycerides (odds ratio with 95% confidence interval per one standard deviation increase in lipid fraction) from polygenic analyses including all lipid-associated variants.

The online version of this article includes the following figure supplement(s) for figure 3:

**Figure supplement 1.** Multivariable Mendelian randomization estimates for HDL-cholesterol, LDL-cholesterol, and triglycerides (odds ratio with 95% confidence interval per one standard deviation increase in lipid fraction) from polygenic analyses including all lipid-associated variants excluding self-reported outcomes.

**Figure supplement 2.** Multivariable Mendelian randomization estimates for HDL-cholesterol, LDL-cholesterol, and triglycerides (odds ratio with 95% confidence interval per one standard deviation increase in lipid fraction) from polygenic analyses including all lipid-associated variants excluding those with a cancer diagnosis other than the site-specific cancer under analysis.

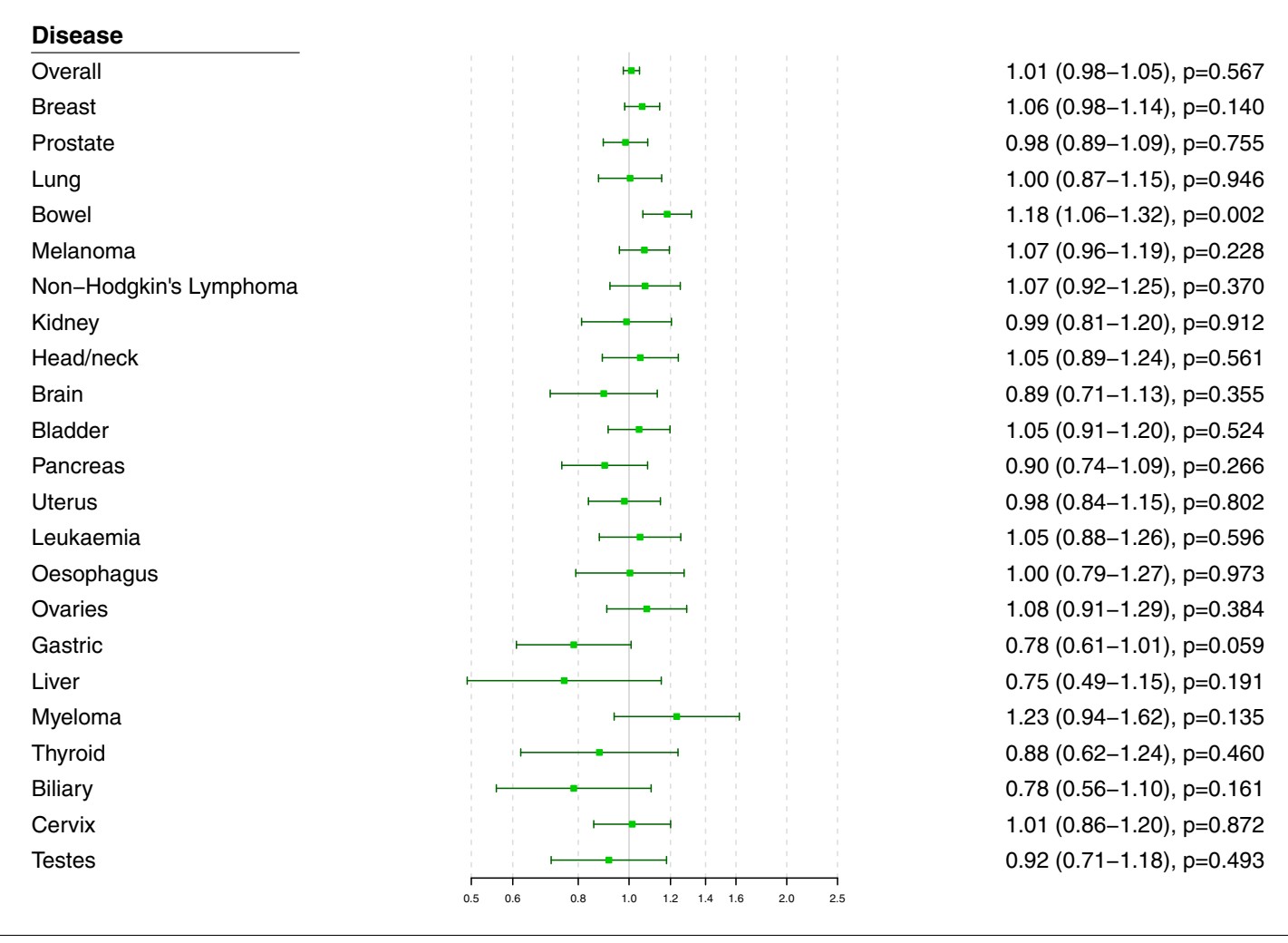

**Figure 4.** Univariable Mendelian randomization estimates for total cholesterol (odds ratio with 95% confidence interval per one standard deviation increase in lipid fraction) from polygenic analyses including all lipid-associated variants.

The online version of this article includes the following figure supplement(s) for figure 4:

**Figure supplement 1.** Univariable Mendelian randomization estimates for total cholesterol (odds ratio with 95% standard deviation increase in lipid fraction) from polygenic analyses including all lipid-associated variants excluding self-reported outcomes.

**Figure supplement 2.** Univariable Mendelian randomization estimates for total cholesterol (odds ratio with 95% confidence interval per one standard deviation increase in lipid fraction) from polygenic analyses including all lipid-associated variants excluding those with a cancer diagnosis other than the site-specific cancer under analysis.

**Figure supplement 3.** Scatterplot to assess heterogeneity of genetic associations with total cholesterol (horizontal axis, standard deviation units) against genetic associations with overall cancer (vertical axis, log odds ratios).

analyses, evidence for a causal effect is most reliable when it is supported by multiple methods that make different assumptions, which was not the case here. No other associations were statistically significant at p<0.01. Again, similar results were observed when omitting self-reported outcomes from the analysis (*Figure 3—figure supplement 1* and *Figure 4—figure supplement 1*), and when omitting individuals with a different cancer diagnosis from the analysis (*Figure 3—figure supplement 2* and *Figure 4—figure supplement 2*). The exception was for liver cancer, where estimates for LDL-cholesterol (OR 0.52, 95% CI 0.30–0.88, p=0.016) and total cholesterol (OR 0.54, 95% CI 0.31–0.94, p=0.028) became stronger on the omission or exclusion of individuals with solely self-reported outcomes. The numbers of events that were self-reported only for each outcome are reported in *Supplementary file 1*. Heterogeneity I (*Crosbie et al., 2013*) statistics were around 40% for the analysis of overall cancer, and generally lower for site-specific cancers (*Supplementary file*

1). The burden of heterogeneity was shared amongst several genetic variants; there were no striking outliers and hence no specific variants strongly driving heterogeneity (see *Figure 4—figure supplement 3* for scatterplot of genetic associations with total cholesterol and overall cancer).

To address the possibility of null results arising due to low power, we combined data on gastrointestinal cancers (liver, stomach, bowel, esophagus, biliary tract, and pancreas). Estimates were somewhat stronger for LDL-cholesterol and total cholesterol compared to the analysis for overall cancer, but did not reach a conventional level of statistical significance: LDL-cholesterol 1.05 (95% CI 0.97–1.14, p=0.23), HDL-cholesterol 1.07 (95% CI 0.97–1.18, p=0.17), triglycerides 1.02 (95% CI 0.91–1.15, p=0.70), and total cholesterol 1.07 (95% CI 0.99–1.16, p=0.10).

## Discussion

Our comprehensive Mendelian randomization investigation shows a positive association between overall cancer and variants in the *HMGCR* gene region which can be considered as proxies for statin therapy. However, gene regions which can be considered as proxies for alternative lipid-lowering therapies were not associated with cancer risk. Furthermore, there was little consistent evidence of an association between genetically-predicted lipid fractions and cancer outcomes in polygenic analyses either for overall cancer or for any site-specific cancer. Taken together, our findings predict that statins lower the risk of cancer, and provide important evidence that this occurs through mechanisms other than lipid lowering.

We found that genetic variants in the *HMGCR* region, serving as proxies for targets of statin therapy, were associated with a 26% decrease in risk of overall cancer per standard deviation (around 39 mg/dL or 1.0 mmol/L) reduction in genetically-predicted LDL-cholesterol. Our result replicates protective associations previously observed for prostate cancer (*PRACTICAL consortium et al., 2016*), colorectal cancer (*Rodriguez-Broadbent et al., 2017*), breast cancer (*Orho-Melander et al., 2018*; *Nowak and Ärnlöv, 2018*), and ovarian cancer (*Yarmolinsky et al., 2020*), although with a stronger weight of statistical evidence due to the additional number of cases analyzed here. For coronary artery disease, the short-term impact of statins in trials is around one-third of the genetic estimate, which represents the impact of lifelong reduced levels of LDL-cholesterol (*Ference et al., 2012*). Under the assumption that even if LDL-cholesterol may not be the relevant causal risk factor, it is a relevant prognostic factor for assessing the degree of HMGCR inhibition, this suggests that any reduction in cancer risk from statins in practice is likely to be modest. Mechanistically, cardiovascular risk reduction by statins is predominantly due to cholesterol lowering (*Liao and Laufs, 2005*), whereas we give evidence this is not the case for cancer.

While our results should be seen as tentative until trials have demonstrated benefit, associations of *HMGCR* variants show broad concordance with statin therapy for many continuous phenotypes (*Würtz et al., 2016*), and suggest that statins reduce the risk of coronary artery disease (*Ference et al., 2016*), increase risk of type 2 diabetes (*Lotta et al., 2016*), and increase risk of intracerebral hemorrhage (*Sun et al., 2019*; *Allara et al., 2019a*), as confirmed in clinical trials (*ASCOT investigators et al., 2003*; *DIAGRAM Consortium et al., 2015*; *Collins et al., 2016*). Genetic evidence pertaining to HMGCR has been proven to be a reliable guide for the performance of statins in trials. Clinical trials are required to confirm our promising findings for primary prevention of cancer risk.

The notion that statins could be used for chemoprevention is longstanding. Nobel Prize winners Goldstein and Brown proposed that this occurs through non-lipid lowering mechanisms (*Goldstein and Brown, 1990*). We provide evidence from human genetics to support this theory. Our results suggest that with respect to genetically predicted HMGCR inhibition and cancer risk, LDL-cholesterol is simply a biomarker of HMGCR inhibition that is accessible, but the true causal pathway is likely via another molecule whose levels are correlated with its LDL-cholesterol lowering effect. HMGCR catalyzes the rate-limiting step of the mevalonate pathway; a pathway with an arm leading to the endpoint of cholesterol synthesis and another arm leading to isoprenoid synthesis. Measuring levels of intra-cellular isoprenoids is challenging but these molecules are implicated in cancer via their role as major post-translational modifiers of key oncogenic proteins (*Mullen et al., 2016*). In particular, mevalonate and other isoprenoid metabolites are required for the prenylation and functioning of the Ras and Rho GTPases, which are oncoproteins, and involved in important cellular processes including apoptosis, phagocytosis, vascular trafficking, cell proliferation,

transmigration, cytoskeleton organization, and recruitment of inflammatory cells. Statin inhibition of these metabolites has demonstrated anti-oncological effects in vivo and in vitro (*Yeganeh et al., 2014*) including the promotion of tumor cell death and apoptosis (*Spampanato et al., 2012*; *Ghosh-Choudhury et al., 2010*; *Parikh et al., 2010*; *Misirkic et al., 2012*), inhibition of angiogenesis (*Park et al., 2008*), and reduction of tumor cell invasion and metastasis (*Wang et al., 2000*; *Yang et al., 2013*). Other potential statin-mediated mechanisms of tumor suppression include the reduction of systemic inflammatory mediators like interleukin 1-beta and tumor necrosis factor (*Park et al., 2008*; *Bruegel et al., 2006*), and epigenetic regulation through inhibiting HMGCR-mediated deacetylation (*Lin et al., 2008*), which contributes to colorectal cancer in mouse models (*Pisanti et al., 2014*). Thus, our findings based on large-scale human genetic data are consistent with pre-clinical studies on statins in cancer which have repeatedly argued for a cholesterol independent mechanism for statin effects on cancer.

We have here presented the first wide-angled Mendelian randomization analysis of lipid subtypes for a range of site-specific cancers. We corroborate previous Mendelian randomization studies suggesting no causal role of lipids in the development of pancreatic cancer (*Carreras-Torres et al., 2017a*) and prostate cancer (*PRACTICAL consortium et al., 2016*). We also extend these findings to show little convincing evidence for total cholesterol levels or lipid fractions as a risk factor for any cancer type studied. However, it is plausible that lack of power or heterogeneity between cancer types could have contributed to these null results as previous large Mendelian randomization studies of the most common cancers (colorectal, lung, and breast) have shown significant associations with LDL-cholesterol, including results in both directions. Although our associations did not achieve a conventional level of statistical significance, we mirror the positive associations of genetically-predicted LDL-cholesterol and HDL-cholesterol with breast cancer (*Nowak and Ärnlöv, 2018*; *Beeghly-Fadiel et al., 2019*) observed in the Breast Cancer Association Consortium which studied over 60,000 cases of cancer. However, we did not observe the negative association observed between LDL-cholesterol and lung cancer found in a Mendelian randomization analysis of 29,266 cases, though in the same study rare LDL-cholesterol variants showed the opposite association (*Carreras-Torres et al., 2017b*). Of all the site-specific cancers studied, the present study only implicated dyslipidemia in driving bowel (i.e. colorectal) cancer with positive associations demonstrated for total and LDL-cholesterol levels. These findings corroborate a previous Mendelian randomization study of 26,397 colorectal cancer patients (*Cornish et al., 2020*). Furthermore, in a smaller Mendelian randomization study, genetically-predicted total cholesterol levels, but not LDL-cholesterol, were associated with colorectal cancer risk (*Rodriguez-Broadbent et al., 2017*), and several previous meta-analyses of observational studies have associated dyslipidemia with increased risk of colorectal adenoma (*Passarelli and Newcomb, 2016*) and cancer (*Yao and Tian, 2015*; *Tian et al., 2015*). However, the associations we found for colorectal cancer were attenuated in sensitivity analyses and must therefore be interpreted with caution. Overall, there was little consistent evidence for total cholesterol levels or lipid fractions as a risk factor for cancer, although this must be interpreted with caution.

Our headline finding relates to overall cancer, which is a combination of different malignancies. While they may have different underlying aetiologies, all cancers are known to share common underlying 'hallmark' molecular and cellular aberrations and there may thus be pathophysiologic relevance in combining outcomes (*Hanahan and Weinberg, 2011*). Furthermore, analyses for overall cancer are highly relevant from a public health perspective. In addition to the cardiovascular benefits of statins, any individual patient decision regarding whether to take them for primary cancer prevention is likely to reflect the risk of all cancer types, not one particular subtype. Overall, clinical trials are needed to confirm the protective effect of statins in the primary prevention of cancer and should characterize the adverse risks of statins before they are advocated in clinical guidance. In particular, any potential adverse effect of LDL-cholesterol lowering on lung cancer should be monitored, despite the lack of replication for this finding in the present analysis.

Our investigation has many strengths, but also limitations. The large sample size of over 360,000 participants and the broad set of outcomes analyzed render this the most comprehensive Mendelian randomization analysis of lipids and cancer outcomes conducted to date. However, the investigation has a number of limitations. For many site-specific cancers, there were not enough outcome events to obtain adequate power to rule out the possibility of moderate causal effects. This is particularly relevant to analyses of gene-specific target regions for LDL-cholesterol, which were not adequately

powered to detect small effect sizes. Conversely, the study of a large number of outcomes across multiple risk factors as we have done is prone to type one errors, meaning that results are falsely found to be significant. However, correction for multiple testing may lead to type two errors, meaning that results are falsely judged to be inconclusive. We encourage readers to weigh the evidence presented carefully rather than to reduce findings to a binary designation of 'significant' or 'not significant'. While there is evidence to support our assumption that genetic variants in relevant gene regions can be used as proxies for pharmacological interventions, our findings should be considered with caution until they have been replicated in clinical trials. Our investigation was able to compare subgroups of the population with different lifelong average levels of lipid fractions, but the impact of lowering a particular lipid fraction in practice is likely to differ from the genetic association, particularly quantitatively (*Burgess et al., 2012*). Combining cancer types to study overall cancer risk has the aforementioned benefits and also results in the largest number of cases and so the greatest power to detect a causal effect. However, this assumes a consistent effect between cancer types, so there is potential for directional heterogeneity and a consequent reduction in power. Furthermore, this combined endpoint is dependent on the characteristics of the analytic sample and the relative prevalence of different cancer types. In particular, cancers with greater survival chances will be over-represented in the case sample. Finally, analyses were conducted in UK-based participants of European ancestries. While it is recommended to have a well-mixed study population for Mendelian randomization to ensure that genetic associations are not influenced by population stratification, it means that results may not be generalizable to other ethnicities or nationalities.

In conclusion, our findings suggest that HMGCR inhibition may have a chemopreventive role in cancer through non-lipid lowering properties and that this role may apply across cancer sites. The efficacy of statins for cancer prevention must be urgently evaluated.

## Materials and methods

### Study design and data sources

We performed two-sample Mendelian randomization analyses, taking genetic associations with risk factors (i.e. serum lipid levels) from one dataset, and genetic associations with cancer outcomes from an independent dataset, as performed previously for cardiovascular diseases (*Allara et al., 2019b*).

We obtained genetic associations with serum lipid concentrations (total cholesterol, LDL-cholesterol, HDL-cholesterol, and triglycerides) from the Global Lipids Genetic Consortium (GLGC) on up to 188,577 individuals of European ancestry (*Global Lipids Genetics Consortium et al., 2013*). Genetic associations were estimated with adjustment for age, sex, and genomic principal components within each participating study after inverse rank quantile normalization of lipid concentrations, and then meta-analyzed across studies.

We estimated genetic associations with cancer outcomes on 367,703 unrelated individuals of European ancestry from the UK Biobank, a population-based cohort recruited between 2006 and 2010 at 22 assessment centers throughout the UK and followed-up until 31st March 2017 or their date of death (recorded until 14th February 2018; *Sudlow et al., 2015*). We defined cancer outcomes for overall cancer and for the 22 most common site-specific cancers in the UK (*Supplementary file 1*). Outcomes were based on electronic health records, hospital episode statistics data, national cancer registry data, and death certification data, which were all coded according to ICD-9 and ICD-10 diagnoses. Further cancer outcomes were captured by self-reported information validated by an interview with a trained nurse and from cancer histology data in the national cancer registry. To obtain genetic association estimates for each outcome, we conducted logistic regression with adjustment for age, sex, and 10 genomic principal components using the *snptest* software program. For sex-specific cancers (breast, uterus, and cervix for women; prostate and testes for men), analyses were restricted to individuals of the relevant sex. For overall cancer, each individual could contribute to the analysis as a case once. For site-specific cancers, an individual could contribute to the analysis of multiple cancers. Controls were defined as individuals without the disease outcome under consideration. Hence an individual with one cancer could be a control for analyses of another cancer. We also performed sensitivity analyses excluding individuals with solely self-reported cancer outcomes from the analyses, and for site-specific cancers, excluding individuals with

a cancer diagnosis other than the site-specific cancer under analysis (so that controls were only those without any cancer diagnosis).

## Gene-specific analyses for HMGCR and other drug proxy variants

We performed targeted analyses for variants in the *HMGCR* gene region that can be considered as proxies for statin therapy. Additionally, we conducted separate analyses for the *PCSK9*, *LDLR*, *NPC1L1*, *APOC3*, and *LPL* gene regions, mimicking other lipid-altering therapies (*Supplementary file 1*). These regions were chosen as they contain variants that explain enough variance in lipids to perform adequately powered analyses. Variants in each gene region explained 0.4% (*HMGCR*), 1.2% (*PCSK9*), 1.0% (*LDLR*), 0.2% (*NPC1L1*), 0.1% (*APOC3*), and <0.1% (*LPL*) of the variance in LDL-cholesterol. The *APOC3* and *LPL* variants also explained 1.0% and 0.9% of the variance in triglycerides, respectively. Variants were chosen based on their associations with the relevant lipid trait from a conditional analysis in the GLGC (Supplementary methods). We performed the inverse-variance weighted method accounting for correlations between the variants using generalized weighted linear regression (*Burgess et al., 2016*). This was implemented using the 'correl' option in the MendelianRandomization package (*Yavorska and Burgess, 2017*). Estimates for the *HMGCR*, *PCSK9*, *LDLR,* and *NPC1L1* gene regions are scaled to a one standard deviation increase in LDL-cholesterol, whereas estimates for the *APOC3* and *LPL* gene regions are scaled to a one standard deviation increase in triglycerides.

## Selection of variants for gene-specific analyses

Variants for the gene-specific analyses were selected to match the choice in a parallel analysis of cardiovascular diseases (*Allara et al., 2019a*). Variants in the *HMGCR* and *PCSK9* regions were originally selected by *Ference et al., 2016*. Variants in the *LPL* region were originally selected by *Lotta et al., 2016*. Variants in the *NPC1L1* region were originally selected by *Ference et al., 2015*. Variants in the *LDLR* and *APOC3* regions were selected by *Do et al., 2013*. All variants were chosen based on their associations with lipid levels in conditional analyses using data from the GLGC. Variants are all conditionally associated with the relevant lipid trait (either LDL-cholesterol or triglycerides) and not strongly correlated ($r^2$ <0.4). The variants are listed in *Supplementary file 1*.

## Polygenic analyses for all lipid-related variants

We carried out polygenic analyses based on 184 genetic variants previously demonstrated to be associated with at least one of total cholesterol, LDL-cholesterol, HDL-cholesterol, or triglycerides at a genome-wide level of significance ($p < 5 \times 10^{-8}$) in the GLGC (*Do et al., 2013*). These variants explained 15.0% of the variance in total cholesterol, 14.6% in LDL-cholesterol, 13.7% in HDL-cholesterol, and 11.7% in triglycerides in the GLGC. Variants were reported as uncorrelated in the original publication by the GLGC, but some pairs of correlated variants remained in the analysis.

To obtain the associations of genetically-predicted values of LDL-cholesterol, HDL-cholesterol, and triglycerides with each cancer outcome while accounting for measured genetic pleiotropy via each other, we performed multivariable Mendelian randomization analyses using the inverse-variance weighted method (*Sanderson et al., 2019*). For total cholesterol, we performed univariable Mendelian randomization analyses using the inverse-variance weighted method (*Burgess et al., 2013*). The analysis for total cholesterol was conducted both because cancer risk may be influenced by total cholesterol rather than any particular lipid subfraction and to mitigate against the loss of power from adjustment for the various lipid subfractions in the multivariable analysis. To account for between-variant heterogeneity, we used random-effects models in all analyses. Heterogeneity between the estimates for different variants was quantified by Cochran's Q statistic and Higgins' I (*Crosbie et al., 2013*) statistic. For polygenic analyses that provided evidence of a causal effect, we additionally performed robust methods for Mendelian randomization, in particular the MR-Egger (*Bowden et al., 2015*) and weighted median methods (*Bowden et al., 2016*). All estimates are expressed per one standard deviation increase in the corresponding lipid fraction (in the GLGC, one standard deviation was 45.6 mg/dL for total cholesterol, 39.0 mg/dL for LDL-cholesterol, 15.8 mg/dL for HDL-cholesterol, and 90.5 mg/dL for triglycerides). Correlation between variants was accounted for in the inverse-variance weighted and MR-Egger methods; for the weighted median analysis, one of each pair of correlated variants was dropped from the analysis.

As power calculators have not been developed for multivariable Mendelian randomization analyses, we performed power calculations for polygenic analyses based on univariable Mendelian randomization for each lipid fraction in turn, and for gene-specific analyses for each gene region in turn (*Burgess, 2014*). We carried out all analyses using R (version 3.4.4) unless otherwise stated. All statistical tests and p-values presented are two sided.

## Acknowledgements

Stephen Burgess is supported by Sir Henry Dale Fellowship jointly funded by the Wellcome Trust and the Royal Society (grant number 204623/Z/16/Z). Stephen Burgess and Amy Mason are supported by the UK National Institute for Health Research Cambridge Biomedical Research Centre.

## Additional information

### Funding

| Funder | Grant reference number | Author |
| --- | --- | --- |
| Wellcome Trust and Royal Society | 204623/Z/16/Z | Stephen Burgess |
| National Institute for Health Research | | Amy M Mason<br>Stephen Burgess |

The funders had no role in study design, data collection and interpretation, or the decision to submit the work for publication.

### Author contributions

Paul Carter, Investigation, Writing - original draft; Mathew Vithayathil, Conceptualization, Writing - original draft; Siddhartha Kar, Rahul Potluri, Susanna C Larsson, Investigation, Writing - review and editing; Amy M Mason, Methodology, Writing - review and editing; Stephen Burgess, Conceptualization, Formal analysis, Methodology, Writing - review and editing

### Author ORCIDs

Stephen Burgess (iD) https://orcid.org/0000-0001-5365-8760

### Decision letter and Author response

Decision letter https://doi.org/10.7554/eLife.57191.sa1
Author response https://doi.org/10.7554/eLife.57191.sa2

## Additional files

### Supplementary files

- Source code 1. R files for performing Mendelian randomization analyses.
- Source data 1. Summarized genetic associations with cancer outcomes estimated in UK Biobank.
- Supplementary file 1. Supplementary tables 1-8.
- Transparent reporting form

### Data availability

All data generated or analysed during this study are publicly-available and/or provided in the supporting files. The UK Biobank can be accessed online at http://biobank.ctsu.ox.ac.uk/crystal/. The Global Lipids Genetics Consortium data can be accessed at http://csg.sph.umich.edu/willer/public/lipids2013/.

The following datasets were generated:

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
