## [Decision Letter]

**Acceptance summary:**

The authors have presented a detailed evaluation of the effects of gene targets of lipid lowering drugs and lipid subfractions on risk of cancer (all-cause and site-specific) using Mendelian randomisation applied to data from the Global Lipid Genetics Consortium and UK Biobank. Importantly, the authors have demonstrated that variants in the HMGCR locus, which are proxies for statin treatment, are associated with all-cause cancer, whilst those in other loci that are proxies for other lipid lowering treatments are not. In contrast, by using genome-wide variation, there was no evidence of a causal effect of lipids on all-cause cancer. The authors conclude that statins may prevent all-cause cancer as well as site-specific cancers through a non-LDL cholesterol mechanism, although they emphasise that the results are not definitive and require further evaluation in clinical trials.

**Decision letter after peer review:**

Thank you for submitting your article "Predicting the effect of statins on cancer risk using genetic variants: a Mendelian randomization study in UK Biobank" for consideration by *eLife*. Your article has been reviewed by three peer reviewers, and the evaluation has been overseen by a Reviewing Editor and a Senior Editor. The reviewers have opted to remain anonymous.

As is customary in *eLife*, the reviewers have discussed their critiques with one another. What follows below is a lightly edited compilation of the essential and ancillary points provided by reviewers in their critiques and in their interaction post-review. Please submit a revised version that addresses these concerns directly. Although we expect that you will address these comments in your response letter, we also need to see the corresponding revision in the text of the manuscript. Some of the reviewers' comments may seem to be simple queries or challenges that do not prompt revisions to the text. Please keep in mind, however, that readers may have the same perspective as the reviewers. Therefore, it is essential that you attempt to amend or expand the text to clarify the narrative accordingly.

Summary:

Carter and colleagues present the results of a Mendelian randomisation study investigating the effects of gene targets of lipid lowering drugs and lipid subfractions on risk of cancer (all-cause and site-specific) using data from the GLGC and UK Biobank. The authors demonstrate the variants in the HMGCR locus (which are proxies for statin treatment) are associated with all-cause cancer – whilst that in other loci that are proxies for other lipid lowering treatments are not. Using genome-wide variation, there was no evidence of a causal effect of lipids on all-cause cancer (but nominal evidence for association with site-specific cancers). The authors conclude that statins may prevent all-cause cancer as well as site-specific cancers through a non-LDL cholesterol mechanism. The authors emphasise that the results are not definitive and require further evaluation in clinical trials.

Overall, the reviewers felt that the manuscript presented a thoughtfully conducted analysis, which was interesting and informative. However, we had some concerns over the interpretation of the results and felt that the manuscript could be strengthened.

Essential revisions:

1) The authors conclude that there was little consistent evidence for associations of LDL cholesterol with site-specific cancers and very weak evidence for an effect on overall cancer risk. This seems to backup the authors inference that the effect of HMGCR on overall cancer is not likely mediated by LDL cholesterol (further supported by results for the other LDL gene targets). However, these results are also compatible with other quite plausible scenarios whereby the null/inconsistent associations reflect highly variable statistical power and heterogeneity across sites.

a) For the most common / most powered cancers from large GWAS consortia (breast cancer, lung cancer and colorectal cancer), MR studies have reported effects of LDL cholesterol. Although the colorectal cancer study cited by the authors reported weak evidence for an effect of LDL cholesterol, a more recent and larger/better powered MR study did report an effect consistent with the authors' result (Cornish et al., 2020).

b) A large MR study of lung cancer risk showed an opposing direction of effect to results for breast and colorectal cancer (Carreras-Torres et al., 2017). Note that the lung cancer study cited by the authors seems to be the wrong one. The results in Figure 2 are also compatible with some directional heterogeneity between cancers. Such heterogeneity would be expected to bias the effect on overall cancer risk towards the null.

c) The HMGCR results and random effects model chosen by the authors imply the presence of horizontal pleiotropy. Did the authors find evidence that pleiotropy was balanced? Unbalanced pleiotropy between SNPs used in the LDL instrument could bias the effect towards the null.

d) As to the LDL gene targets, the null results for *PCSK9*, *LDLR* and *NPC1L1*, especially for *NPC1L1*, are also compatible with a lack of power for small effect sizes. In addition, bias from measurement error in the SNP-LDL effect could also attenuate results towards the null. The larger effect of HMGCR on cancer risk compared to other targets could reflect the summed effects of LDL and non-LDL related pathways.

e) It should also be pointed out that few published MR studies of cancer seem to mutually adjust for lipid subfractions, which seems to increase the effect sizes in comparison to the univariate analysis, which would reduce power of those studies.

2) The authors definition of all-cause cancer likely includes a substantial proportion of non-melanoma skin cancers, most of which are likely to be relatively benign basal cell carcinomas. The authors perform sensitivity analyses for the HMGCR association, but not for analyses of the lipid subfractions, which are important given the evidence for prostate cancer that the effect of LDL cholesterol is stronger for more aggressive cancers. The definition also includes self-reported cases, and it would be useful to present sensitivity analyses excluding these cases for the site-specific cancers. It was also not clear if individuals diagnosed with multiple cancers contribute to multiple analyses, or it was only the first cancer diagnosis that was counted. Further clarity regarding the control group would be helpful (i.e. was there a common cancer-free control group used for all analyses or could an individual diagnose with one cancer be used as a control for another cancer).

3) How were the variants in each gene region selected? Were these all variants that reached a pre-defined threshold in GLGC, that were subsequently LD pruned? If so, further details are required in the Materials and methods section. Would it be more appropriate to run an approximate conditional analysis using the GLGC summary statistics? Would it be more appropriate to use effect sizes (for both exposure and outcome) that were derived from a joint model (and not univariate models)? The genetic variants have been selected in specific gene regions to be proxies for different lipid-lowering treatments. However, if the selection is based only on association with lipids, can the authors be sure that they are operating through the genes that are the targets of the drugs. Are all the selected variants missense variants or have strong support as eQTLs of the relevant genes? Some further details would be very useful.

4) For the polygenic analyses, are the 184 variants LD pruned? If so, further details are provided in the Materials and methods section. It would be useful to clarify why total cholesterol has been consider by itself in a separate univariable MR, whilst LDL, HDL and TG are considered together in a multivariable MR. If the same variants are used in both sets of analyses, is this optimal (or appropriate)? Random effects models were appropriately used to account for between-variant heterogeneity. Was there indeed significant heterogeneity between variants?

5) The authors write that the association with bowel cancer for LDL/total cholesterol was attenuated in MR-Egger/weighted median analyses, but is this a fair comparison given that these sensitivity analyses did not take into account pleiotropy with other lipid fractions (as in the multivariable MR)? This is particularly important since it feeds into the general narrative about LDL cholesterol not being causally relevant in cancer.

6) Throughout, it is not entirely clear what threshold is being used for significance. It was not clear that associations would meet a multiple testing correction for the number of site-specific cancers.

7) It would be useful to have some discussion of the following issues:

a) The authors argue that from a public health and primary care point of view, overall cancer risk is more important. This is a fair point but is undermined by the possibility of directional heterogeneity amongst cancer sites. The effect on lung cancer (reported by a separate much larger MR study) is especially concerning, given the very poor survival rates for this cancer.

b) The authors state that power should be greatest in the analysis combining all-cause cancer, but this assumes homogenous effects across site-specific cancers, and thus could be lower in the presence of heterogeneity (especially directional heterogeneity).

c) Based on MR/trial results for statins and coronary disease, the authors write that reductions in cancer risk through statin treatment are likely to be modest. A possible problem with this interpretation is that the mechanism of the effect on coronary disease is through LDL cholesterol whilst that with cancer may not be. Does this interpretation require an assumption that the SNP-LDL effect is strongly correlated with the non-LDL/pleiotropic effect? In addition, the effect of statins could be greater if the effect of statins occurs through both LDL and non-LDL mechanisms, which seems possible for some site-specific cancers (e.g. colorectal cancer and breast cancer).

[Editors' note: further revisions were suggested prior to acceptance, as described below.]

Thank you for submitting your revised article "Predicting the effect of statins on cancer risk using genetic variants from a Mendelian randomization study in UK Biobank" for consideration by *eLife*. Your article has been reviewed by a Senior Editor, a Reviewing Editor, and two reviewers.

We are happy to see the effort you made at amending the paper to accommodate the concerns and suggestions from the reviewers. In this new review round, the reviewers have discussed the reviews with one another and the Reviewing Editor has drafted this decision to help you prepare a new revised submission.

Summary:

The authors have made changes to the manuscript to address concerns raised by the reviewers, which they believe has improved the manuscript. However, concerns remain, particularly with the use of individuals with a given cancer as a control for another cancer.

Essential revisions:

The authors write that an individual with cancer could serve as a control for another cancer. "Hence an individual with one cancer could be a control for analyses of another cancer." This seems incongruent with the authors' hypothesis that cholesterol could be a common cause for different cancers. Wouldn't this be expected to reduce power for site-specific cancers? For example, if you include a colorectal cancer case as a control for an analysis of breast cancer, won't this attenuate the effect on breast cancer to the null (assuming LDL causes both cancers, which seems to be supported by previous MR studies in BCAC, CORECT/GECCO)?

---

## [Author Response]

Essential revisions:1) The authors conclude that there was little consistent evidence for associations of LDL cholesterol with site-specific cancers and very weak evidence for an effect on overall cancer risk. This seems to backup the authors inference that the effect of HMGCR on overall cancer is not likely mediated by LDL cholesterol (further supported by results for the other LDL gene targets). However, these results are also compatible with other quite plausible scenarios whereby the null/inconsistent associations reflect highly variable statistical power and heterogeneity across sites.a) For the most common / most powered cancers from large GWAS consortia (breast cancer, lung cancer and colorectal cancer), MR studies have reported effects of LDL cholesterol. Although the colorectal cancer study cited by the authors reported weak evidence for an effect of LDL cholesterol, a more recent and larger/better powered MR study did report an effect consistent with the authors' result (Cornish et al., 2020).b) A large MR study of lung cancer risk showed an opposing direction of effect to results for breast and colorectal cancer (Carreras-Torres et al., 2017). Note that the lung cancer study cited by the authors seems to be the wrong one. The results in Figure 2 are also compatible with some directional heterogeneity between cancers. Such heterogeneity would be expected to bias the effect on overall cancer risk towards the null.c) The HMGCR results and random effects model chosen by the authors imply the presence of horizontal pleiotropy. Did the authors find evidence that pleiotropy was balanced? Unbalanced pleiotropy between SNPs used in the LDL instrument could bias the effect towards the null.d) As to the LDL gene targets, the null results for PCSK9, LDLR and NPC1L1, especially for NPC1L1, are also compatible with a lack of power for small effect sizes. In addition, bias from measurement error in the SNP-LDL effect could also attenuate results towards the null. The larger effect of HMGCR on cancer risk compared to other targets could reflect the summed effects of LDL and non-LDL related pathways.e) It should also be pointed out that few published MR studies of cancer seem to mutually adjust for lipid subfractions, which seems to increase the effect sizes in comparison to the univariate analysis, which would reduce power of those studies.

Thank you for these suggestions. These are important alternative explanations of the results which require further discussion.

a) We thank the reviewers for this additional reference (Cornish et al.), which we have added to the manuscript.

b) Thank you – we intended to cite this paper, but mistakenly cited the wrong lung cancer paper by Carreras-Torres.

To address this question of low power and heterogeneity, we have combined data on gastrointestinal cancers into a single analysis. The motivation for this is that several cancers for which some evidence of a positive effect of LDL-cholesterol have been observed are gastrointestinal cancers (colorectal cancer as noted above, reported as bowel cancer in our analysis), and it is plausible that a cholesterol-rich diet increases the risk of various gastrointestinal cancers (for example, via increased exposure to bile acids). Combining evidence across related outcomes can improve power. Estimates from multivariable Mendelian randomization are: LDL-cholesterol OR 1.05 (95% CI 0.97, 1.14, p=0.23), HDL-cholesterol OR 1.07 (95% CI 0.97, 1.18, p=0.17), triglycerides 1.02 (95% CI 0.91, 1.15, p=0.70). Estimates from univariable Mendelian randomization are: total cholesterol OR 1.07 (95% CI 0.99, 1.16, p=0.10).

c) It is possible that unbalanced pleiotropy may bias an estimate towards the null, but there is no reason why unbalanced pleiotropy would be more likely to bias results towards the null rather than away from the null. Unfortunately, it is not possible to determine the direction of pleiotropy in a Mendelian randomization analysis definitively.

If the multivariable InSIDE assumption holds, then the average pleiotropic effect is estimated by the intercept term in a multivariable MR-Egger method. The intercept from the multivariable MR-Egger method applied to the analysis of total cancer (variants orientated to LDL-cholesterol increasing allele) is 0.0013 (SE 0.0008, p = 0.12). Estimates for the lipid fractions remain null using this method: LDL-cholesterol OR 0.99 (95% CI 0.95, 1.04, p=0.71), HDL-cholesterol OR 0.98 (95% CI 0.94, 1.03, p=0.44), triglycerides 0.98 (95% CI 0.93, 1.04, p=0.53). We have added this result to the manuscript (Table 5 in Supplementary file 1).

d) We have now commented on the potential for lack of power again when interpreting the lipid results, “We also extend these findings to show little convincing evidence for total cholesterol levels or lipid fractions as a risk factor for any cancer type studied. However, it is plausible that lack of power or heterogeneity between cancer types could have contributed to these null results as previous large MR studies of the most common cancers have shown significant associations with LDL-cholesterol, including results in both directions.” However, it is worth noting that the same genetic variants had statistically robust associations with coronary artery disease and other cardiovascular diseases despite the limited power.

e) While accounting for multiple exposures may reduce power as variants explain less of the variability in the exposures, effect estimates in a multivariable Mendelian randomization analysis may be larger than in a univariable Mendelian randomization analysis, which would increase power. However, the choice of primary analysis should be determined first based on the plausibility of assumptions, and only secondarily to maximize statistical power. As the genetic variants are associated with multiple lipid fractions, a multivariable Mendelian randomization analysis is recommended. This analysis allows us to include a large number of genetic variants in the analysis, which results in improved power. To address the issue of reduced power (amongst other reasons), we additionally performed analyses for total cholesterol using univariable Mendelian randomization (i.e. with no adjustment for alternative risk factors). Any potential loss of power due to having multiple exposures does not apply to this analysis.

2) The authors definition of all-cause cancer likely includes a substantial proportion of non-melanoma skin cancers, most of which are likely to be relatively benign basal cell carcinomas. The authors perform sensitivity analyses for the HMGCR association, but not for analyses of the lipid subfractions, which are important given the evidence for prostate cancer that the effect of LDL cholesterol is stronger for more aggressive cancers. The definition also includes self-reported cases, and it would be useful to present sensitivity analyses excluding these cases for the site-specific cancers. It was also not clear if individuals diagnosed with multiple cancers contribute to multiple analyses, or it was only the first cancer diagnosis that was counted. Further clarity regarding the control group would be helpful (i.e. was there a common cancer-free control group used for all analyses or could an individual diagnose with one cancer be used as a control for another cancer).

We have added sensitivity analyses for the polygenic analysis for overall cancer excluding self-reported events, and restricting to the 22 site-specific cancers described in the paper. Results for these analyses are given below:

There is some evidence that the effect of LDL-cholesterol is stronger when restricting the analysis to the 22 site-specific cancers, although the evidence for LDL-cholesterol as a causal risk factor is still not statistically robust.

We have also performed analyses excluding self-reported outcomes for each of the 22 site-specific cancers (Figure 2—figure supplement 7 and Figure 2—figure supplement 8). Results are largely similar to the main analyses which included self-reported outcomes, with no associations found with the majority of site-specific cancers. These analyses replicated the positive associations found in main analyses between bowel cancer and LDL-cholesterol (OR 1.16 , 95% CI 1.04-1.29) and total cholesterol (OR 1.18, 95% CI 1.05-1.31). The inverse association between liver cancer and LDL-cholesterol was replicated and that with total cholesterol was strengthened (OR 0.52, 95% CI 0.30-0.88). Similarly, the positive association between triglycerides and melanoma risk was replicated (OR 1.22, 95% CI 1.04-1.44), and, the negative association between triglycerides and lung cancer risk was strengthened (OR 0.82, 95% CI 0.68-0.99). These results therefore support the main analyses which included self-reported outcomes.

An individual could contribute to the analysis of multiple cancers. However, for overall cancer, each individual could only contribute as a case event once.

Controls were defined as individuals without the disease outcome under consideration. Hence an individual with one cancer could be a control for analyses of another cancer. These points are now clarified in the manuscript.

3) How were the variants in each gene region selected? Were these all variants that reached a pre-defined threshold in GLGC, that were subsequently LD pruned? If so, further details are required in the Materials and methods section. Would it be more appropriate to run an approximate conditional analysis using the GLGC summary statistics? Would it be more appropriate to use effect sizes (for both exposure and outcome) that were derived from a joint model (and not univariate models)? The genetic variants have been selected in specific gene regions to be proxies for different lipid-lowering treatments. However, if the selection is based only on association with lipids, can the authors be sure that they are operating through the genes that are the targets of the drugs. Are all the selected variants missense variants or have strong support as eQTLs of the relevant genes? Some further details would be very useful.

For the gene-specific analyses, our choice of variants was the same as previous papers investigating these gene regions (in particular, Allara et al.). We would strongly prefer to keep the same choice of variants to allow direct comparisons with this previous work.

The following details have been added to the manuscript: “Variants in the HMGCR and PCSK9 regions were originally reported by Ference et al. (2013). Variants in the LPL region were originally reported by Lotta et al. (2016). Variants in the NPC1L1 region were originally reported by Ference et al. Variants in the LDLR and APOC3 regions were reported by Do et al. (2013). All variants were chosen based on their associations with lipid levels in conditional analyses using data from the GLGC. Variants are all conditionally associated with the relevant lipid trait (either LDL-cholesterol or triglycerides) and not strongly correlated (r^2^<0.4).”

Correlations between variants were accounted for in the analysis model. The inverse-variance weighted method for correlated variants takes as inputs the marginal genetic associations with the exposure and outcome from univariate regression analyses (Burgess et al., 2016).

Analyses based on each gene region have shown positive estimates for coronary artery disease in Mendelian randomization analyses, consistent with the known effect of LDL-cholesterol and the putative effect of triglycerides on coronary artery disease risk. This is a positive control that supports the choice of the genetic variants in the present analysis.

It is noted that the objective of Mendelian randomization is not to test for causality with respect to a genetic variant (or genetic variants), but to test for causality with respect to a phenotypic risk factor. Therefore, it is not necessary that a variant used in a Mendelian randomization investigation is itself a causal variant; it may be in linkage disequilibrium with a causal variant. All that is required is that the variant divides the population into groups that differ with respect to their average levels of the risk factor.

4) For the polygenic analyses, are the 184 variants LD pruned? If so, further details are provided in the Materials and methods section. It would be useful to clarify why total cholesterol has been consider by itself in a separate univariable MR, whilst LDL, HDL and TG are considered together in a multivariable MR. If the same variants are used in both sets of analyses, is this optimal (or appropriate)? Random effects models were appropriately used to account for between-variant heterogeneity. Was there indeed significant heterogeneity between variants?

The 184 variants are taken from the analyses of Do et al., and were also previously using in Allara et al. The variants were supposedly pruned to independence by the original authors, but some pairs of correlated variants remained. Correlation between these variants was accounted for in all analyses.

Total cholesterol was additionally considered as an exposure as it is plausible that it is a risk factor for cancer. The analysis of total cholesterol includes all genetic variants and so potentially has greater power than the multivariable analysis of the lipid subtypes (see point 1). Performing a univariable analysis also allows a greater range of robust methods to be performed. While we acknowledge that there are potential criticisms of this analysis, there are also potential strengths. Our preference is to present a range of analyses, and allow the reader to judge the strength of evidence.

We now present the degree of heterogeneity in the multivariable analyses of the lipid fractions and the univariable analyses of total cholesterol (Table 7 in Supplementary file 1). The degree of heterogeneity was relatively low to moderate across analyses.

5) The authors write that the association with bowel cancer for LDL/total cholesterol was attenuated in MR-Egger/weighted median analyses, but is this a fair comparison given that these sensitivity analyses did not take into account pleiotropy with other lipid fractions (as in the multivariable MR)? This is particularly important since it feeds into the general narrative about LDL cholesterol not being causally relevant in cancer.

Evidence for a risk factor being a cause of the outcome is most reliable when it is supported by multiple methods that make different assumptions – this is the principle of triangulation of evidence. Again, analyses for the causal effect of LDL-cholesterol on cardiovascular diseases are generally robust across multiple methods. While there are reasons why the MR-Egger and weighted median analyses can be criticized, the same is true for all of the analysis methods. We have edited the text to clarify that these analyses are univariable analyses, but we maintain that they add additional evidence to address the question of causality.

6) Throughout, it is not entirely clear what threshold is being used for significance. It was not clear that associations would meet a multiple testing correction for the number of site-specific cancers.

We agree that care must be taken when interpreting significant results in analyses such as this one which have a large number of outcomes across multiple risk factors. For this reason, we performed sensitivity analyses for the two positive results we found in polygenic analyses for site-specific cancers: “Results were attenuated in robust methods (Table 7 in Supplementary file 1).” We did not detect robust associations between total cholesterol or lipid subtypes and site-specific cancers in this study, and this formed part of the basis for our overall conclusion. However, we have made the potential limitation of limited power and multiple testing clearer by adding, “Conversely, the study of a large number of outcomes across multiple risk factors as we have done is prone to type 1 errors, meaning that results are falsely found to be significant. However, correction for multiple testing may lead to type 2 errors, meaning that results are falsely judged to be inconclusive. We encourage readers to weigh the evidence presented carefully rather than to reduce findings to a binary designation of ‘significant’ or ‘not significant’.”

If we were overly strict in accounting for multiple testing, then our analyses could be criticized for lack of power. If we did not account for multiple testing, then we could be criticized for promoting false positive findings. Particularly given the large number of supplementary and sensitivity analyses, and the fact that a binary designation of results as “significant” or “non-significant” is unhelpful, we prefer to present results in a way that allows the reader to judge the strength of evidence rather than to impose our own judgement.

7) It would be useful to have some discussion of the following issues:a) The authors argue that from a public health and primary care point of view, overall cancer risk is more important. This is a fair point but is undermined by the possibility of directional heterogeneity amongst cancer sites. The effect on lung cancer (reported by a separate much larger MR study) is especially concerning, given the very poor survival rates for this cancer.

Thank you for raising this point. We have moved our discussion of the public health message of the overall cancer result into a separate paragraph in order to emphasise it (Discussion section). As part of this paragraph we have also discussed the potential for detrimental effects on individual cancer sites that may be of clinical importance, despite this overall cancer result. We used the lung cancer result as an important example of this.

b) The authors state that power should be greatest in the analysis combining all-cause cancer, but this assumes homogenous effects across site-specific cancers, and thus could be lower in the presence of heterogeneity (especially directional heterogeneity).

This is an important limitation and we have now directly addressed it in the limitations paragraph.

c) Based on MR/trial results for statins and coronary disease, the authors write that reductions in cancer risk through statin treatment are likely to be modest. A possible problem with this interpretation is that the mechanism of the effect on coronary disease is through LDL cholesterol whilst that with cancer may not be. Does this interpretation require an assumption that the SNP-LDL effect is strongly correlated with the non-LDL/pleiotropic effect? In addition, the effect of statins could be greater if the effect of statins occurs through both LDL and non-LDL mechanisms, which seems possible for some site-specific cancers (e.g. colorectal cancer and breast cancer).

The true effect size is difficult to estimate and could only be accurately assessed using a clinical trial. However, *HMGCR* variants can be considered as a measurable proxy for inhibition of the HMGCR pathway, and so the association of these variants with cancer risk would include both the cholesterol and non-cholesterol lowering effects. As such, the results for cancer risk inform regarding the totality of the effect of inhibition of the HMGCR pathway, and should provide an accurate indication of the potential effect of statins, however that operates. This analysis is agnostic to the mechanism; our interpretation of non-LDL mechanisms playing a role comes from the results of the paper as a whole with no associations seen in other gene regions and in LDL-cholesterol analyses.

[Editors' note: further revisions were suggested prior to acceptance, as described below.]

The authors have made changes to the manuscript to address concerns raised by the reviewers, which they believe has improved the manuscript. However, concerns remain, particularly with the use of individuals with a given cancer as a control for another cancer.

We thank the reviewers for their careful reading of the manuscript and comments. We have addressed the reviewers’ concerns by providing additional figure supplements excluding non-target cancer cases from the analysis (rather than treating them as controls). The reason for not changing the main presentation of results is to respect the original statistical analysis plan. If this is a non-negotiable point for the reviewers, then we would be willing to change this decision and alter the main presentation of results in the manuscript – this would be a simple swap of which figures are main figures and which figures are supplementary figures. However, we are reluctant to deviate from the pre-specified statistical analysis plan for the main presentation of results unless it is clear that the original analysis was incorrect, which does not appear to be the case here.

Essential revisions:The authors write that an individual with cancer could serve as a control for another cancer. "Hence an individual with one cancer could be a control for analyses of another cancer." This seems incongruent with the authors' hypothesis that cholesterol could be a common cause for different cancers. Wouldn't this be expected to reduce power for site-specific cancers? For example, if you include a colorectal cancer case as a control for an analysis of breast cancer, won't this attenuate the effect on breast cancer to the null (assuming LDL causes both cancers, which seems to be supported by previous MR studies in BCAC, CORECT/GECCO)?

We appreciate the reviewers’ concern. However, it is not clear that there is a right or wrong answer here – this appears to be a question of preference. If we view the optimal analyses of an observational study as imitating a randomized trial (the “target trial” perspective: see Hernán and Robins, 2016, Am J Epidemiol.), then it would be unusual for the analysis of a randomized trial for one disease to exclude from analysis participants who developed an alternative disease during follow up. So, while we take the reviewers’ point that this may decision may have a conservative impact on power, we do not think it is an incorrect analysis choice. We are therefore reluctant to deviate from the pre-specified statistical analysis plan for our main analyses.

We have re-calculated genetic associations with outcomes omitting individuals with any other cancer diagnosis from the analysis, rather than treating them as controls. This is treated as an additional supplementary analysis. Results for the main analyses of the paper are presented in Figure 2—figure supplement 8 (this mirrors Figure 2B from the manuscript), Figure 3—figure supplement 2 (this mirrors Figure 3 from the manuscript), and Figure 4—figure supplement 2 (this mirrors Figure 4 from the manuscript). Estimates for HMGCR are higher in this analysis for all outcomes. Estimates for lipid subfractions and total cholesterol are similar to those from the main analyses, and findings are the same in each set of analyses.

We have also re-calculated genetic associations for analyses excluding self-reported cancer cases excluding non-target cancer cases from the analysis. Updated results are presented in Figure 3—figure supplement 1 and Figure 4—figure supplement 1. We have also added estimates for the HMGCR gene region excluding self-reported outcomes (Figure 2—figure supplement 7). Again, results are almost identical, and findings are unchanged.